# Modeling the Critical Success Factors for BIM Implementation in Developing Countries: Sampling the Turkish AEC Industry

Seda Tan, Gulden Gumusburun Ayalp *, Muhammed Zubeyr Tel, Merve Serter and Yusuf Berkay Metinal

Department of Architecture, Hasan Kalyoncu University, Gaziantep 270000, Turkey;
seda.tan@std.hku.edu.tr (S.T.); mzubeyr.tel@hku.edu.tr (M.Z.T.); merve.serter1@std.hku.edu.tr (M.S.);
yberkay.metinal@std.hku.edu.tr (Y.B.M.)
* Correspondence: gulden.ayalp@hku.edu.tr; Tel.: +90-342-211-8080

**Abstract:** One of the latest advancements transforming the global architectural, engineering, and construction (AEC) industry is building information modeling (BIM). Although BIM implementation is at high level in developed countries, it is at a lower level in developing countries. BIM is new to the construction industry in Turkey, with only minor construction firms having implemented it. When making projections based on the current state of the Turkish AEC industry, it is foreseen that it will become mandatory in the near future. Considering this projection, it is doubtful that many construction companies will be caught unprepared for this situation and will not know how to implement BIM. Therefore, this study aimed to identify and model the critical success factors for BIM implementation and their impact size in order to gain insight for the fast and efficient implementation of BIM among construction firms in the Turkish AEC industry, which can be generalized for most developing countries. To reach these aims, a questionnaire was designed with 41 identified success criteria (SC) that were derived through a systematic literature review (SLR). The survey was conducted on construction professionals who actively implement BIM technology at their occupied firms in Turkey and they were asked to rank the importance of 41 SC on a five-point Likert scale. The sampling frame consisted of architects and civil engineers, and in total, 243 responses were received. The differentiation between SC and critical success criteria (CSC) was obtained by using a normalized mean cutoff value. An exploratory factor analysis (EFA) was used to identify the critical success factors (CSFs), and structural equation modeling (SEM) was used to examine the underlying size effects of each CSF on BIM implementation in the Turkish AEC industry. The results of this study reveal 20 CSC for successful BIM implementation, and EFA exhibited three CSFs from 20 CSC. Three critical success factors for BIM implementation in the Turkish construction industry were determined and grouped into two categories. "Awareness of technological benefits" and "organizational readiness and competitive advantages" formed one group and are the most influential critical success factors for BIM implementation. "Motivation of management regarding BIM" formed the second group of critical success factors that have a significant effect.

**Keywords:** building information modelling; critical success factors; BIM implementation; structural equation modelling

## 1. Introduction

Completing every construction project within the scheduled time and expected budget and achieving high quality are significant in the architectural, engineering, and construction (AEC) industry. Time lapses as a result of not realizing these targets in the planned direction, which result in an increase in cost, are undesirable for an efficient construction process. Therefore, the importance of construction project management is increasing. However, in recent years, besides efficient construction processes, project life cycle processes, including the use/operation process of the building, have gained importance. In addition, sustainability has come to the fore in this process. Sustainability is a major criterion in the AEC industry.

Girginkaya et al. [1] presented that building information modeling (BIM) users tended to have a higher awareness of sustainability. At this point, BIM emerges as a revolutionary technological innovation in the AEC industry with its multidimensional approach.

BIM is a technology that enables the use of digital information models [2] for better quality and efficient construction management in the design, construction, and operational processes [3] throughout the life cycle of a project. Digital transformation and sustainability have garnered increased attention in the AEC industry in recent years. This domain's primary technologies include BIM, virtual design and construction (VDC), and integrated digital delivery (IDD) [3]. Among all these concepts, the use of BIM in sustainable construction improves project efficiency and productivity; enables real-time sustainable design and multi-design possibilities; simplifies the selection of sustainable materials; and decreases energy consumption, material waste, and the environmental effect of a project [4]. Therefore, many professionals regard BIM as a significant opportunity in the AEC industry since it is a revolutionary technology and process [5] for sustainability. BIM has emerged as a solution to facilitate the integration and management of information throughout the building life cycle, thereby providing an opportunity for making the best use of the available design data for sustainable design and performance analysis [5,6].

As BIM allows multidisciplinary information to be layered within a single model, this approach provides an opportunity for sustainability improvement measures and environmental performance analyses to be performed precisely and efficiently [7]. Within this scope, various functions of BIM exist that contribute to sustainability, such as energy performance simulation, the assessment of building sustainability performance, lighting analysis, and construction and demolition waste analysis [5]. With all these features, BIM provides an integrated, interactive, and virtual approach to underpin building design, construction, and operation [3].

Owing to the aforementioned advantages, BIM awareness is high in most of the developed countries around the world (The United States of America, The United Kingdom, France, Finland, South Korea, etc.). Governments in developed countries have primarily followed the BIM imperative to encourage the implementation of BIM in public projects. Accordingly, the first BIM guidelines for public projects were announced in 2007 by Nordic countries, such as Norway, Denmark, and Finland. Subsequently, standards within specific frameworks for the implementation of BIM were published by developed countries (The United States, the United Kingdom, Norway, Denmark, Finland, and Germany). The use of BIM has been made mandatory in various states and public institutions in line with certain limits (project cost, size, etc.) [8]. Thus, to date, over 15 countries around the world have announced their various plans for or have mandated BIM [9].

In the Turkish AEC industry, BIM implementation became mandatory in public infrastructure and transportation projects and is enforced by the Turkish government. Furthermore, "Project Management Procedures" and "BIM Technical Specifications" have been prepared by the Republic of Turkey's Ministry of Transport and Infrastructure to be used in transportation projects construction tenders to increase efficiency in information management within the scope of digital management targets [10].

Turkey has a strong and growing economy according to its gross domestic product (GDP). When the data of developed and developing countries are examined, it can be determined that there is a strong relationship between the construction sector and GDP [11]. The construction industry has a driving role in the economy of Turkey. According to 2020 data, the direct contribution of this industry accounted for 5.4% of the total GDP, and contributed approximately 30% together with other sectors [12]. According to the "Top 250 International Contractors List" of Engineering News-Record (ENR) in 2022, Turkey ranked third with 40 contractor businesses, just after the United States (41) [13]. To remain competitive, the Turkish AEC industry needs to closely follow the technological developments ongoing in the world and adapt various strategies to the conditions of the country [14]. At this point, the use of BIM technology is of critical importance for the necessary development and progress in the competitive environment of the construction industry [15].

Despite the high level of experience and the good reputation of Turkish contractors in the construction industry and especially in international contracting projects [16], Turkish contractors need to work on increasing their competitiveness and corporate performance in international markets in order to increase their income [17]. Therefore, BIM has become even more crucial for them to increase the number and size of their projects by providing engineering, procurement, and construction services. To address this challenge, it is critical for the construction industry to identify the critical success factors of BIM implementation in order to better understand the process, and to suggest strategic proposals.

Previous studies have shown the factors that influence the success of BIM implementation and adoption, including management support and organizational readiness [8,18–30], technical requirements [21–23,31,32], client acceptance/alignment [25,28,30], human resources and the people dimension [23,30,32,33], collaboration and coordination of project partners [18,19,28,30,31,33], and process and change management [30–34].

In a brief review of the relevant research, several success factors/enablers are identified, all of which are important; however, it remains unclear as to which of these are the most effective factors, and the nature of the relationships between them. For this reason, it is not possible to make strategic plans or paths for successful BIM implementation and adoption in the first stage. It is expected that the obligation to implement BIM in infrastructure and transportation projects enforced by the public will also be mandated for superstructure/upper structure projects in the near future in Turkey. Therefore, it is vital that BIM adoption is fast and efficient in organizations that have not adopted BIM yet when BIM implementation becomes mandatory. From this perspective, knowing the key success factors in BIM implementation is very important when making a strategic plan for the successful implementation and extensive use of BIM. In other words, there is a literature gap and a lack of knowledge about critical success factors for BIM implementation from the perspective of BIM user stakeholders.

To fill this gap in the literature, the main objectives of the current research are fourfold: (1) To determine the success factors of BIM implementation. (2) To identify the critical success factors of those identified. (3) To model the identified critical success factors, in order to determine the importance of each factor in the construction industry. This model can help to implement the actions necessary to successfully implement and increase the prevalence of BIM. Finally, (4) this study addresses the gap in current knowledge by identifying key success factors for BIM implementation in the Turkish AEC, which can be generalized to almost all developing countries. Although BIM implementation is uncommon and the adoption of BIM is lagging in Turkey, this study investigates the critical success factors during the implementation of BIM by collecting the opinions of architects and engineers who use BIM via a survey questionnaire, unlike previous studies. In addition, the present study investigates the original critical success factors for BIM implementation while considering the opinions of minority construction BIM practitioners. These aspects constitute an important difference from previous studies.

## 2. Research Background and Gaps in the Literature

### 2.1. Research Background

The drivers/influencers of successful BIM implementation and adoption in the construction industry could be categorized into four stages by examining pioneering studies: BIM practices [21–23,27,32,35], BIM awareness [27,28,36,37], organization [21–23,27,28,34,35,38,39], and BIM education [19,28,38,40].

#### 2.1.1. BIM Practices

The construction industry is distinguishable from other industries, such as the manufacturing industry, in terms of its sector practices and its complex structure, which includes many participants/stakeholders and actions. This circumstance causes a decrease in productivity and efficiency in construction practices [41]. The implementation of BIM as a technological innovation [42] and the integration of all project stakeholders and their

behaviors/attitudes towards BIM and processes are important to increase construction efficiency [43]. In this respect, BIM practices are significant for rapid BIM adoption and more successful BIM implementation.

### 2.1.2. BIM Awareness

The behaviors relating to adopting innovation are shaped depending on the features of the group or groups expected to adopt the innovation and environmental influences. Along with this, there is a direct, positive relationship between the level of awareness and adoption. Therefore, it is significant that the relative advantages [44] and tangible benefits of BIM technology should be known by the industry stakeholders for its adoption and implementation in the construction industry. It can be stated that BIM awareness plays a considerable role in successful BIM implementation via reducing stakeholders' resistance to change [45].

### 2.1.3. BIM Organization

In the acceptance and adoption of technological innovation, potential benefits and advantages in the competitive environment are considered. Technological adoption could be at the national level, but it is usually at the level of organizational acceptance, depending on top management decisions and organizational strategy [46]. The successful implementation of a technological innovation requires the adoption of changes by the administrative components of the organization [47], thus creating organizational motivation. Hence, it could be stated that organizational influences are important to successful BIM implementation.

### 2.1.4. BIM Education

Education is necessary for the successful implementation of change. In particular, employees/participants need appropriate change-related training [48]. Investments in education are significant in the AEC industry, in order to realize the potential of BIM and to profit from all of the benefits of BIM [49]. However, people are at the center of change management for technological adoptions. It could be stated that education is an important factor determining the strength of the link of between humans and technology. As a result, BIM education-related influences play an important role in the success of BIM implementation.

### 2.2. Gaps in the Literature

Although BIM technology is a new research subject, several noteworthy BIM-related studies have been conducted from different perspectives, including BIM adoption and implementation [26,41,50–56], maturity levels [57], BIM adoption and implementation challenges and barriers [24,39,41,51,52,58–63], drivers [35,36,64–67], the benefits of BIM implementation [37,68] from different perspectives and practices [40,69], motivations [70], and critical risk factors [2,71]. Furthermore, some studies have investigated the critical success factors/key enablers for different perspectives of BIM, such as lean practices [33], facility management [72,73], construction phases [27], level 2 implementation [28], the contractual framework [74,75], the delivery of BIM projects [76], maturity models [46], e-negotiation practices [77], and the impact of BIM drivers and awareness on project life cycles [65].

Studies have also been conducted to investigate the critical success factors/enablers of BIM implementation and adoption in developing countries [8,19–23,25,34,78,79] and in developed countries [18,29,38]. To gain deeper insight into previous studies regarding the identification of the CSFs of BIM implementation, it is significant to note that some of these studies performed literature reviews [18,34] and included studies published from 2005 to 2015 and from 2004 to 2019, respectively. The remaining studies on the CSFs of BIM implementation adopted quantitative methods in which they reviewed the previous studies without a systemized foundation, identified the critical success factors [8,19,21,23,25], and then surveyed these factors with questionnaires. Furthermore, they analyzed the obtained data's descriptive statics, using ranking analyses [21,25] and exploratory fac-

tor analyses [8,19,22,23]. Unlike these studies, Al-Ashmori et al. [78] adopted qualitative semi-structured interviews and structured questionnaires. Similarly, Abbasnejad et al. [29] identified the key enablers of BIM implementation with expert interviews and modelled them with interpretive structural modeling.

Several studies reviewed the literature to compose the questionnaire; however, a systematic review has not been conducted. Only a few studies have used systematic reviews [20] for organizing a structured questionnaire. In addition, when the above studies were analyzed, it was observed that the number of studies focusing on the CSFs of BIM implementation conducted with the structural equation model (SEM) is limited.

The literature contains a number of studies on BIM implementation in different regions, such as South Korea [43,80], the Czech Republic [81], Taiwan [21,82], the United Kingdom [28,30,83], Sweden [59], Saudi Arabia [54], Algerian [55], Ghana [84], Croatia [85], Nigeria [86], India [87], Vietnam [74,88], Slovakia [76], and Singapore [8]. However, the number of studies on the use of BIM technology in the Turkish AEC industry is relatively limited compared to other countries. It is thought that this is because BIM implementation has not been mandated in construction projects carried out by both public and private companies in the Turkish AEC industry.

The contents and methodologies of previous BIM-related studies conducted in the Turkish AEC industry are as follows. Tan and Gumusburun Ayalp [89] identified root factors limiting BIM implementation in Turkey using an exploratory factor analysis, a confirmatory factor analysis, and structural equation modeling, and then presented a structural model. Elmalı and Bayram [90] investigated the sectoral awareness level of BIM implementation in the Turkish AEC industry via descriptive statistical methods and a ranking analysis, and looked into how to overcome the obstacles of BIM adoption. Challenges to BIM adoption in mega construction projects were determined using descriptive and ranking analyses in the study of Akcay [91]. Simsek and Uzun [92] revealed that condominium and land valuation could be realized with the created 3D BIM model via the development of a mathematical process. Tezel et al. [93] examined the relationship between FM and BIM in the context of Turkey with descriptive statistics using data obtained via quantitative data collection. Demirdöğen et al. [94] focused on the solutions to problems in FM from a lean management philosophy, and developed an FM framework for healthcare facilities with a design science research methodology. Aydın and Oral [95] investigated the effect of evaluating design parameters with BIM Revit software in determining the energy efficiency of the settlement texture. Çakır and Uzun [96] examined the effects of the use of BIM-based software on the design abilities of architecture students. Erpay and Sertyesilisik [97] aimed to prevent possible legal problems in the BIM project life cycle at the contract phase by developing a preliminary checklist. Sarı and Pekeriçli [98] compared and evaluated the official BIM documents with the Turkish architecture and engineering service specification document by examined the application guides published in the United States and the United Kingdom. Toklu and Mayuk [99] evaluated current BIM usage in Turkey and compared it with implementations in countries where the usage of BIM is high. Furthermore, Yilmaz et al. [100] proposed a reference model to evaluate the BIM capability of AEC/FM processes by examining eight BIM capability and maturity models determined in the literature. Alshorafa and Ergen [101] determined the LOD for BIM implementation in large-scale projects that were carried out in Turkey, Qatar, and Saudi Arabia. Koseoglu et al. [102] investigated the relationship between BIM and lean practices through the gains achieved in the Istanbul Grand Airport project. Koseoglu and Nurtan-Gunes [103] examined lean interactions related to mobile BIM processes through a framework by focusing on digital transformations of construction sites. Aladag et al. [50] examined the driving forces and obstacles of BIM implementation in the Turkish construction industry using focus group discussions and analyzed their responses with a basic ranking technique. Ademci and Gundes [104] researched the drivers and barriers to BIM adoption and implementation at the individual and organizational levels in Turkey using a questionnaire survey and analyzed the descriptive statistics and hypothesis tests.

Although the findings of existing studies on BIM implementation are significant, there is a lack of identification of critical success factors in the Turkish AEC industry.

It is observed that existing studies have used descriptive statistics, rank analyses, and explanatory factor analyses to study the challenges to BIM adoption or identify critical success factors. However, no study has determined the critical success factors of BIM implementation and their impact size on common BIM either in Turkey or developing countries. In addition, no study has focused on the key success factors of BIM implementation using SEM in developing countries. Therefore, there is a gap regarding key success factors in the literature on BIM research in developing countries.

This study highlights the critical success factors of BIM implementation and their impact size specific to the Turkish AEC industry, which can be generalized to all developing countries. In this regard, the success criteria (SC) are derived from the literature through a systematic literature review (SLR). To determine the critical success criteria (CSC), a normalized mean value analysis was conducted. Then, exploratory factor analyses were performed to determine the factors. Subsequently, a confirmatory factor analysis was applied to validate the latent factors, and SEM was conducted to establish the critical success factors (CSFs) of BIM implementation. The use of SEM allows a deeper understanding of the underlying relationships between the CSFs and their impacts with the path coefficients in this study.

Similarly to previous studies, this study employed a survey. Unlike previous studies, the questions of the survey were composed by utilizing an SLR, which provides more reliable and objective criteria. Therefore, in contrast to previous studies, deeper quantitative statistical methods were used to identify the critical success factors of BIM implementation, and a model was developed to determine the impact size of each CSFs on BIM implementation.

## 3. Materials and Methods

The comprehensive methodological approach of this study is presented in Figure 1. The research began with an SLR to determine the success criteria (SC) that support BIM implementation. This was followed by the organization and validation of the questionnaire. Data collection began with an online survey of the participants. A data analysis was initiated with a reliability analysis of the obtained data. Then, critical success criteria (CSC) were identified using the normalized mean value ranking. Following this, an exploratory factor analysis (EFA) was then performed to identify the latent factors, which involved factor extraction and factor rotation. Subsequently, based on the determined latent factors, SEM was constructed to identify the critical success factors for BIM implementation. The statistical calculations were performed using Statistical Package for the Social Sciences (IBM SPSS) 22.0, and LISREL 8.7.

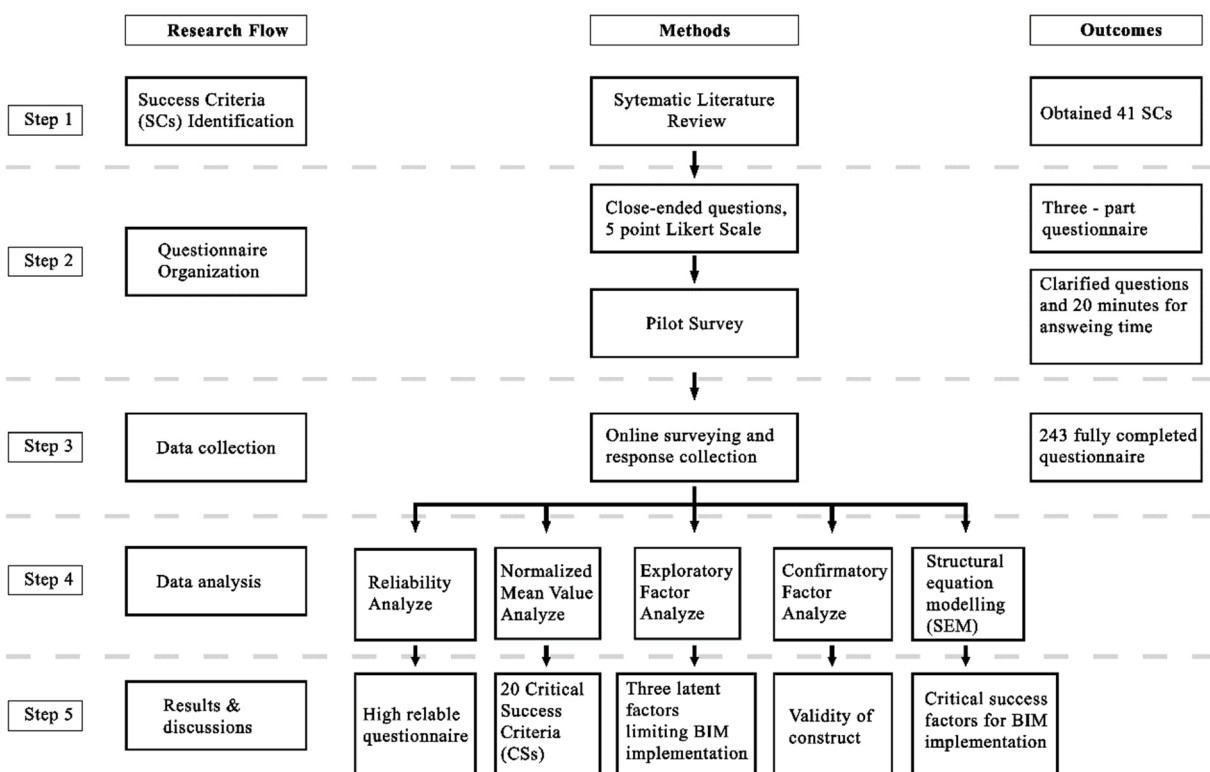

**Figure 1.** Research methodology framework.

### 3.1. Identifying the Success Criteria (SC)

The first stage of this study involved the determination of the SC that support the use of BIM for all pre-construction and post-construction stages related to BIM practices, BIM awareness, BIM organization, and BIM education. In conducting an SLR, the extensive literature was examined. SLR is a method-driven, transparent, and reproducible method [105] for analyzing and understanding all research related to a certain issue, subject, or phenomenon [106]. An SLR decreases prejudice by performing extensive literature searches in relation to published and unpublished research and also provides a record of reviewers' decisions, processes, and findings [107]. Unlike other methods, such as citation-based approaches, SLR is a powerful tool for evaluating published work in the scientific field. In the present study, the renowned three-stage approach was used to identify causes. It involved the "planning the review", "conducting the review", and "documenting the review" stages, which are presented in Figure 2.

The research question was designated at the planning stage and a review protocol was developed. First, primary studies were identified in the transitive review phase. Subsequently, the identified studies were selected, extracted, analyzed, and synthesized. Finally, the outputs obtained from the literature were published as a report during the examination and documentation phase.

Web of Science (WoS) search engines were used to find scientific papers regarding BIM implementation since this database contains nearly all major research articles and includes built-in analytic capabilities for producing representative numbers. WoS was considered for its accuracy, comprehensiveness, and coverage of several study fields [108]. The search was limited to articles and review papers in English published in academic journals between 2012 and 2022.

Keywords were used to set search parameters in the WoS database. The full search code was "All Fields", and the search parameters were as the following keywords: "BIM application" AND "construction" AND "success factors", NOT "road", NOT "highway", NOT "infrastructure".

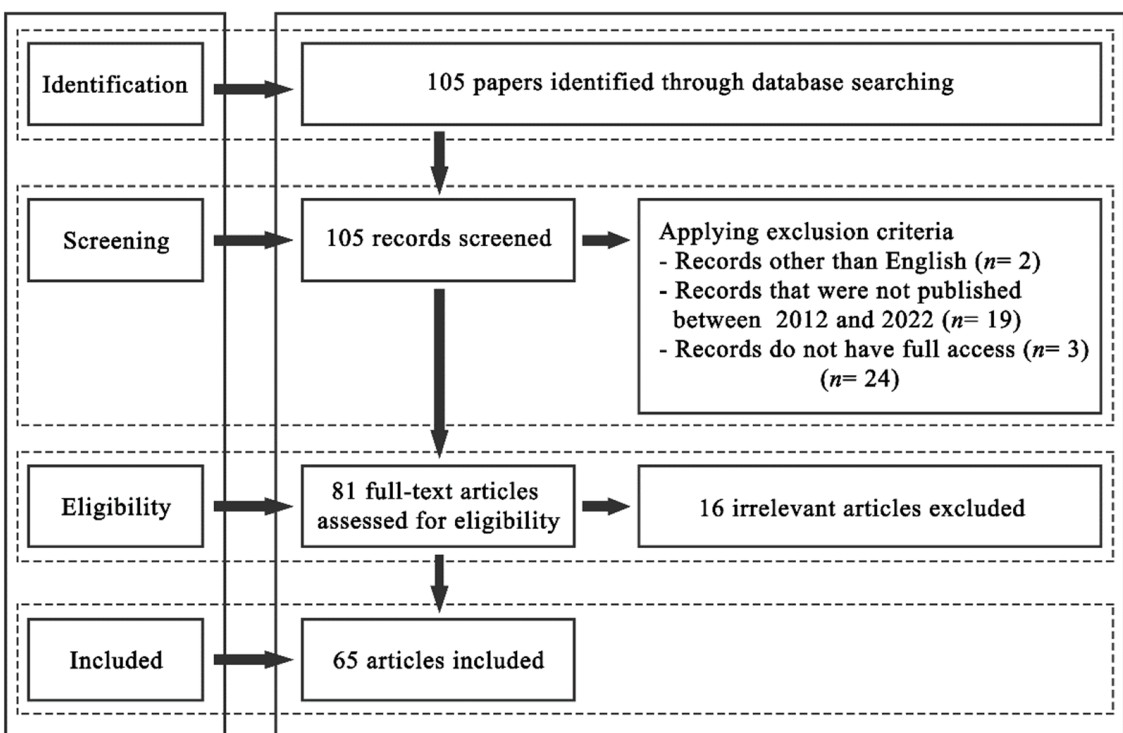

**Figure 2.** Stages of systematic literature review.

In total, 105 journal articles were identified during the initial search. It is recommended that at least two authors work independently to assess studies that meet the review's defined inclusion and exclusion criteria [109]. Therefore, the findings were confirmed by a crosscheck among the authors to determine their adherence to our criteria. Based on the study topic, researchers should define inclusion and exclusion criteria [110]. Among the 105 records, 24 of them were removed after applying the exclusion criteria.

In the second stage, 81 full-text studies published in English between 2012 and 2022 with full access available were finally considered relevant and therefore included. Due to irrelevancy, 16 studies were excluded. Finally, 65 studies were selected for thorough examination to identify SC. The 65 selected articles were coded in order to relate the key study findings to particular categories of causes, taking into consideration BIM practices, BIM awareness, BIM organization, and BIM education. Eventually, 41 SC that were directly related to these four categories were determined and are listed in Table 1.

**Table 1.** Success criteria (SC) for BIM implementation derived from SLR.

| Main Category | SC Coded as | Success Criteria | Sources |
|---|---|---|---|
| BIM Practices | P1 | BIM technology reduces cost and time | [18,24,26,33,34,37,41,46,56,60,65,67,69, 70,75,79,84,85,111–122] |
| | P2 | Workflow, productivity, and efficiency are unaffected by the transition to BIM | [4,6,19,37,41,56,60,72,114,117,123] |
| | P3 | Stakeholders use BIM technology | [6,19,21,25,69,79,85,114,121,124] |
| | P4 | BIM provides knowledge sharing between stakeholders | [18,19,21,24,25,29,33,34,37,38,41,53,56, 66,67,75,79,88,116,117,125–129] |
| | P5 | Creating more efficient projects with the participation, coordination, and supervision of the stakeholders | [6,21,23,28,33,34,37,53,56,65,67,69,78, 119,121,124,126,130] |
| | P6 | Ease of learning in BIM-based programs | [37,66,114,128] |

**Table 1.** *Cont.*

| Main Category | SC Coded as | Success Criteria | Sources |
|---|---|---|---|
| BIM Awareness | A1 | Accessibility of design, schedule, and budget data during the design stage with BIM | [6,8,38,60,114,128] |
| | A2 | Availability of quality, schedule, and cost information during construction with BIM | [21,37,38,56,128,131] |
| | A3 | Availability of performance, usability, and financial information at the management stage with BIM | [6,38,56,69,127,128,131] |
| | A4 | Possibility of alternative design options analysis and simulation with BIM | [8,18,21,33,41,60,65,69,70,79,111,120, 128,132] |
| | A5 | With BIM, an increase in work quality and adaptation to the planned time can be achieved, and an accurate quantity and cost estimation can be made with the created building model | [18,21,24,26,33,37,56,67,69,70,75,113, 119,120,126,128,131] |
| | A6 | Clash detection capabilities of BIM among projects | [8,18,21,24,33,34,37,41,56,65,69,70,88, 114,117,119,120,128,131,133,134] |
| | A7 | The BIM allows control of all systems in a 3D model, with instant and automatic intervention | [8,18,21,33,53,75,78,112,117,128] |
| | A8 | BIM standardizes information and supports a variety of file formats | [6,8,21,53,85,86,114,117,128,135] |
| | A9 | Potential failures, leaks, and evacuation plans can be graphically illustrated and adapted with BIM | [21,26,33,46,65,70,128] |
| BIM Organization | O1 | Supporting the use of BIM by top and middle management | [25,28,29,120,128] |
| | O2 | Inclusion of BIM in the competitive environment of the industry | [8,21,24,41,46,65,66,69,117,119] |
| | O3 | Presence of qualified BIM personnel | [6,21,23,25,28,88,114,117,128] |
| | O4 | Sufficiency of financial resources of organization for high initial investment costs in the transition to BIM | [21,28,46,66,69,75,114,117,120,124,127, 128,132,133,135,136] |
| | O5 | Availability of return on investment | [19,41,65,66,70,79,119,127] |
| | O6 | BIM adoption requires individual and group motivation in the organization | [6,19–21,28,29,32,78,79,117,127,128] |
| | O7 | The ease of adoption by personnel of BIM technology | [6,19,21,25,38,53,75,114,137] |
| | O8 | Knowledge and demands of clients about BIM technology | [6,8,21,23,25,28,66,83,84,112,117,122, 128,135] |
| | O9 | Less risk in projects prepared with BIM technology | [18,19,21,33,67,69,75,78] |
| | O10 | Adaptation of the stakeholders involved in the construction, inspection, and use processes, starting from the design process to the implementation of BIM technology | [19,20,33,38,69,78,114,128] |
| | O11 | The need for significant changes in the organizational structure for integration with BIM technology (size, structure, culture of the organization type, etc.) | [8,19,21,23,25,34,53,75,117,127] |
| | O12 | Formation of a young and dynamic team, with the new business model causing a change in the decision mechanism and workload distribution | [19,88,121,125,128] |
| | O13 | Starting the implementation immediately after training in the transition process to BIM | [86] |
| | O14 | Having the necessary information and technological infrastructure for BIM applications within the institution | [6,19,21,23,25,53,78,86] |

**Table 1.** *Cont.*

| Main Category | SC Coded as | Success Criteria | Sources |
|---|---|---|---|
| BIM Organization | O15 | The financial structure of the institution is sufficient for BIM technology | [19,23,25,53,69,78,127] |
| | O16 | Receiving consultancy on BIM technology by universities and specialist companies | [19,23,25,75,128] |
| | O17 | Possibility for the existing in-house environment to learn and practice something new | [19,21,25,128] |
| | O18 | Ability to learn and practice something new in an in-house environment | [8,21,23,34,78,79,81,117,125,128] |
| | O19 | Allowing information sharing in seminars, workshops and conferences on BIM technology organized by different companies | [23,84,86] |
| BIM education l | E1 | Having courses on BIM technology in the education programs of the architecture and engineering departments of universities | [85,86,128] |
| | E2 | Availability of faculty members who are knowledgeable about BIM technology in universities | [8,20,75,128] |
| | E3 | Ability to conduct studies, albeit new, to understand the differences in the design process of the traditional methods and BIM technology of the professional chambers related to the construction sector in Turkey | [20,29,86,114,128] |
| | E4 | Introducing the transition process to BIM technology for companies by professional chambers related to the construction sector in Turkey | [6,8,20,21,23,28,53,75,86,114,128,135] |
| | E5 | Access to resources in mother tongues about BIM technology | [6,28,29,53,135] |
| | E6 | Having sectoral or academic training opportunities related to BIM in the city of residence | [25,29,33,34,86] |
| | E7 | Equipping the personnel working in the organization with training programs, seminars, knowledge, and skills in line with their needs | [6,8,19–21,23,28,30,33,38,53,75,83,86, 88,114,116,117,127,128,135] |

*3.2. Organizing the Questionnaire*

Based on the SLR, a questionnaire was organized and administered to architects and engineers working in architectural offices and construction companies in Turkey that use BIM technology. The questionnaire comprised three main parts. To measure all of the SC supporting the implementation of BIM, the first part of the questionnaire comprises four fragments that can be summarized as related to BIM practices, BIM awareness, organization and BIM education, and demographic variables. The survey comprised 41 SC (6 for practices, 9 for awareness, 19 for organization, and 7 for education) that were to be assessed on a five-point Likert-type scale, with 1 representing "not severe" and 5 representing "most severe". All participants rated the importance of the 41 SC affecting BIM use according to their presumptions and expectations.

The second part contained one question that was included to gain insights into the participants' general perception of the level of BIM implementation in the Turkish AEC industry using a five-point Likert scale.

The participants' personal and sociodemographic information was obtained in the third part, which comprised questions focusing on profession, gender, education, experience in the construction industry, and occupation.

A pilot study to identify confusing or not fully understood statements was conducted to ensure that the statements in the questionnaire were clear. Additionally, the response time for the questionnaire was determined. The pilot study involved a total of 20 participants (10 from each profession) with more than five years' experience in the field. The final questionnaire was revised using the comments and suggestions from the pilot study.

### 3.3. Data Collection

The target population of this research included Turkish architects and engineers. The sample group contained architects and engineers that worked in companies that use BIM technology.

The final questionnaire was distributed via e-mail to 1028 architects and civil engineers on 21 February 2022 using a random sampling method. Responses were accepted until 14 May 2022, and a total of 257 questionnaires were returned, of which 14 were removed because of missing data, and 243 completed questionnaires were used as material for the current study, representing a response rate of 23.63%. Akintoye [138] suggested that an acceptable response rate for construction research is between 20% and 30%. Therefore, this response rate was considered to be acceptable.

The population number in this research is extracted from the statistics produced by the Turkish Chamber of Engineers and Architects Union for 2022 as 205,843 [139]; the number architects and civil engineers are 68,478 and 137,365, respectively. The random sampling technique is widely used in construction research, in which the sample is randomly selected from the population based on a non-zero probability [140]. This technique is considered to be effective because it produces a sampling representative of the population by avoiding any voluntary response bias [141]. All individuals in a population have an equal chance of being selected as the sample and providing an accurate representation for the broader population [142]. Therefore, this technique is adopted to select the participants for this study. The method to determine the calculation of the sample size of population is adopted from Enshassi and Al Swaity [143] as follows:

$$\text{SS} = \frac{P(1-P) \times Z^2}{C^2} \tag{1}$$

where SS = Sample size; $Z$ = Z value (1.96 for 95 percent confidence level); $P$ = percentage picking a choice, expressed as a decimal (0.5 used for sample size needed); and $C$ = margin of error (9 percent), in which the maximum error of estimation which can be 9 or 8 percent.

$$\text{SS} = \frac{0.5(1-0.5) \times 1.96^2}{0.09} = 118.57 \approx 119 \tag{2}$$

To check the marginal error value, the following formula is used from Enshassi and Al Swaity [143]:

The maximum margin of error for a 95 percent confidence level $\approx \dfrac{1.96}{\sqrt{SS}} = \dfrac{1.96}{119} = 0.18 > 0.09$. The margin is acceptable and the minimum size is 119; therefore, the obtained 243 data are deemed acceptable as well. In addition, sample size is an important issue for structural equation modelling because it relates to the stability of the parameter estimates [144].

When the sample size was evaluated for adequacy for SEM, the existing literature suggested that sample sizes between 100 and 400 are sufficient for structural equation modeling analysis [145]. According to Iacobucci [146], minimum and maximum sample sizes of 50 and 100, respectively, can be sufficient, and the rule of thumb suggesting a required sample size of at least 200 is 'conservative' and 'simplistic'. Additionally, Hair et al. [147] suggested that the appropriate sample size for SEM should be a minimum of 200 and a maximum of 400. Within this scope, the sample size of 243 in the present study can be considered sufficient.

### 3.4. Data Analysis

The participants' responses were coded and analyzed using the Statistical Package for Social Sciences (SPSS) 22.0 and LISREL 8.7 software to perform several statistical tests, such as a reliability analysis, a normalized mean value analysis, an exploratory factor analysis (EFA), a confirmatory factor analysis (CFA), and structural equation modeling (SEM).

To determine the internal consistency among questions using a Likert scale in a survey, the reliability should be measured [148]. Cronbach's alpha ($\alpha$) was utilized to determine the statistical reliability and validity of the participants' replies. The $\alpha$ coefficient ranged from "0" to "1"; the minimum acceptable reliability threshold was determined as 0.7 [149].

To identify the critical success criteria among the identified SC, a normalized mean values (NMVs) analysis was conducted for each of the 41 SC. The calculation of NMVs for each success criteria follows Equation (1). Any SC with an NMV exceeding 0.5 is considered to meet the critical success criteria (CSC). Liao and Teo [8], Xu et al. [150], and Zhao et al. [151,152] used this method of ranking analysis to classify CSC. Additionally, to further support the selection of identified CSC, the mean values of the individual CSC were checked to see if they exceeded the total mean value of the success criteria. Any success criterion with a mean value exceeding the total mean value of all success criteria was considered to be a CSC. This method was used by Won et al. [80] and Liao and Teo [8] to support the selection of CSC based on ranking analyses.

$$Normalized\ mean\ value = \frac{(mean\ of\ success\ criteria - lowest\ ranked\ mean)}{(highest\ ranked\ mean - lowest\ ranked\ mean)} \tag{3}$$

To achieve one of the main objectives of this study, it is important to identify the underlying factor structure. To highlight the critical factors, the responses to the CSC contained in the questionnaire were imported into the SPSS program and subjected to EFA using varimax rotation (eigen value = 1 as the cut-off). Accordingly, the main factors were identified as ICs, with a factor loading greater than 0.5 [148].

Following the EFA analysis, a confirmatory factor analysis (CFA) was performed on all the identified ICs using the LISREL software to create construct validity. Construct validity refers to how successfully a hypothesized factor has been quantified [153] and survey questions with higher construct validity can better assess the characteristics they claim to reveal. The CFA was utilized as the primary indicator of validity. Multiple fit indices were selected to demonstrate evidence of a good fit between the model and data, including the chi-square ($\chi^2$) test statistic, comparative fit index (CFI), and root mean square error of approximation (RMSEA). In a CFA, the path coefficients among variables are referred to as effect sizes, which have values of less than 0.1 for small effects, around 0.3 for medium effects, and greater than or equal to 0.5 for large effects [154]. In this study, associations with path coefficients of 0.5 or greater and t-values of more than 2.58 were considered significant (99% confidence level).

Finally, an SEM was developed using LISREL 8.7 to quantitatively identify the root factors limiting BIM implementation. This analysis provides an opportunity to confirm the sufficiency of the model concerning the relationship between measurement paths and latent variables. While there are several outlooks for the adequacy of path coefficients above the 0.1 threshold, a path coefficient of 0.2 is recommended [155]. The higher the path coefficient, the stronger the relationship between the independent and dependent constructs of a path [154].

## 4. Results

### 4.1. Reliability and Validity of Questionnaire

The Cronbach's $\alpha$ coefficient of the dataset for the 41 success criteria (SC) regarding BIM implementation and the level of usage of BIM technology in the Turkish construction industry was determined to be 0.971, which is above the minimum threshold of 0.7 [149]. The $\alpha$ coefficients of each SC are presented at Table 2.

**Table 2.** Ranking and selection of CSC (*n* = 243).

| Code of Success Criteria | Cronbach's Alpha | Mean | Standard Deviation | Normalized Mean | Rank |
|---|---|---|---|---|---|
| P1 | 0.972 | 3.46 | 1.147 | 0.62 * | 11 |
| P2 | 0.971 | 3.16 | 1.001 | 0.36 | 30 |
| P3 | 0.971 | 3.23 | 1.336 | 0.42 | 28 |
| P4 | 0.972 | 3.31 | 1.360 | 0.49 | 22 |
| P5 | 0.971 | 3.64 | 1.160 | 0.78 * | 4 |
| P6 | 0.971 | 3.43 | 1.135 | 0.60 * | 14 |
| A1 | 0.970 | 3.22 | 1.240 | 0.41 | 29 |
| A2 | 0.970 | 3.45 | 1.123 | 0.61 * | 13 |
| A3 | 0.970 | 3.30 | 1.193 | 0.48 | 23 |
| A4 | 0.970 | 3.50 | 1.189 | 0.66 * | 7 |
| A5 | 0.970 | 3.64 | 1.191 | 0.78 * | 5 |
| A6 | 0.971 | 3.65 | 1.169 | 0.79 * | 3 |
| A7 | 0.971 | 3.89 | 1.044 | 1.00 * | 1 |
| A8 | 0.970 | 3.69 | 1.098 | 0.83 * | 2 |
| A9 | 0.970 | 3.49 | 1.190 | 0.65 * | 10 |
| O1 | 0.971 | 3.14 | 1.186 | 0.34 | 32 |
| O2 | 0.971 | 3.46 | 1.189 | 0.62 * | 12 |
| O3 | 0.971 | 3.25 | 1.194 | 0.44 | 25 |
| O4 | 0.970 | 3.25 | 1.341 | 0.44 | 27 |
| O5 | 0.970 | 3.25 | 1.264 | 0.44 | 26 |
| O6 | 0.970 | 3.28 | 1.259 | 0.46 | 24 |
| O7 | 0.971 | 3.16 | 1.085 | 0.36 | 31 |
| O8 | 0.971 | 3.44 | 1.316 | 0.60 * | 16 |
| O9 | 0.970 | 3.36 | 1.253 | 0.53 * | 20 |
| O10 | 0.970 | 3.51 | 1.261 | 0.66 * | 8 |
| O11 | 0.970 | 3.49 | 1.157 | 0.65 * | 9 |
| O12 | 0.970 | 3.43 | 1.175 | 0.60 * | 15 |
| O13 | 0.970 | 3.31 | 1.226 | 0.49 | 21 |
| O14 | 0.970 | 3.40 | 1.247 | 0.57 * | 17 |
| O15 | 0.971 | 3.38 | 1.203 | 0.55 * | 19 |
| O16 | 0.971 | 3.05 | 1.354 | 0.26 | 34 |
| O17 | 0.971 | 3.35 | 1.126 | 0.53 * | 18 |
| O18 | 0.970 | 3.62 | 1.109 | 0.76 * | 6 |
| O19 | 0.970 | 3.14 | 1.247 | 0.34 | 33 |
| E1 | 0.972 | 2.98 | 1.317 | 0.20 | 35 |
| E2 | 0.972 | 2.78 | 1.345 | 0.02 | 40 |
| E3 | 0.971 | 2.88 | 1.234 | 0.11 | 39 |
| E4 | 0.971 | 2.90 | 1.185 | 0.13 | 36 |
| E5 | 0.971 | 2.75 | 1.194 | 0.00 | 41 |
| E6 | 0.971 | 2.89 | 1.199 | 0.12 | 37 |
| E7 | 0.971 | 2.89 | 1.220 | 0.12 | 38 |

* Denotes critical success criteria (CSC).

*4.2. Ranking Analysis*

The ranking analysis revealed 20 critical success criteria (CSC) out of the 41 SC, which can be listed in descending order as follows: A7, A8, A6, P5, A5, O18, O10, A4, A9, O11, O2, P1, A2, O8, P6, O12, O14, O15, O9, and O17. The means and standard deviations of the initial 41 SC were calculated and are tabulated in Table 2.

The success criteria with the highest mean value (i.e., A7, mean = 3.89) was given the rank of 1. The success criteria with the lowest mean value (i.e., E5, mean = 2.75) was given the rank of 41 (Table 2). The ranking analysis indicated that 20 out the 41 SC obtained NMVs exceeding 0.5 and these were considered to be critical success criteria (CSC). Additionally, to further support the selection of these 20 CSC, the mean values of each CSC were found to have exceeded the total mean value of all the success criteria, which was calculated as 3.30; therefore, 20 CSC were obtained from the ranking analysis.

*4.3. Designating Critical Success Factors Affecting BIM Implementation—EFA*

The factor structure is significant because an elementary aim of this study is to determine the critical success factors affecting BIM implementation. Toward this aim, an exploratory factor analysis (EFA) was performed to identify the latent factors. In this study, the EFA was extracted using the principal component method, and the Kaiser normalization of the varimax rotation was applied. CSC with a loading factor greater than 0.5 were determined to be the primary success factors. Table 3 summarizes the 20 CSC, along with their factor loadings.

**Table 3.** Results of EFA and CFA.

| Factors | Code of CSC | Exploratory Factor Analyze | | | Confirmatory Factor Analyze |
|---|---|---|---|---|---|
| | | *Eigen Value* | *% of Variance* | *Factor Loadings* | *Standardized Coefficients* |
| Awareness of technological benefits (ATB) | A6 | | | 0.839 | 0.84 |
| | A7 | | | 0.837 | 0.81 |
| | A9 | | | 0.784 | 0.83 |
| | A5 | | | 0.755 | 0.86 |
| | A8 | 11.451 | 30.314 | 0.722 | 0.79 |
| | A2 | | | 0.683 | 0.83 |
| | A4 | | | 0.646 | 0.80 |
| | O9 | | | 0.641 | 0.81 |
| | P5 | | | 0.512 | 0.66 |
| Organizational readiness, and competitive ad-vantages (ORCA) | O14 | | | 0.820 | 0.86 |
| | O8 | | | 0.772 | 0.70 |
| | O15 | | | 0.745 | 0.77 |
| | O12 | | | 0.744 | 0.88 |
| | O17 | 1.561 | 29.649 | 0.706 | 0.67 |
| | O18 | | | 0.703 | 0.77 |
| | O11 | | | 0.695 | 0.87 |
| | O10 | | | 0.632 | 0.85 |
| | O2 | | | 0.591 | 0.68 |
| Motivation of management regarding BIM (MMB) | P1 | 1.261 | 11.404 | 0.847 | 0.73 |
| | P6 | | | 0.838 | 0.83 |
| **Total explained variance** | | 71.367 | | $\chi^2$**/df** | 2.90 |
| **Kaiser–Meyer–Olkin (KMO) value** | | 0.891 | | **RMSEA** | 0.04 |
| **Barlett's test of sphericity** | Approx. chi-square | 4660.242 | | **CFI** | 0.96 |
| | df | 190 | | **GFI** | 0.97 |
| | *p* | 0.000 | | **AGFI** | 0.92 |

The sample adequacy value for Kaiser–Meyer–Olkin (KMO) is 0.891, which is greater than 0.5, indicating that the sampling was adequate for factor analysis [156]. Bartlett's test of the data reported a significant value of $\chi^2$ (4660.24), *p* < 0.000, indicating that correlations between items were sufficient to perform EFA. The loadings of CSC into their corresponding groups were then revealed using a varimax rotation, as indicated in Table 3. All component loadings exceeded 0.50, which suggests that all CSC loaded onto their respective factors are significant [157].

Each factor was assigned a name corresponding to the nature of the latent factors which load onto that particular component. The interpretations, labels, and abbreviations for each of these components are as follows:

Factor 1: Awareness of technological benefits (ATB);
Factor 2: Organizational readiness, and competitive advantages (ORCA);
Factor 3: Motivation of management regarding BIM (MMB).

### 4.4. Confirmatory Factor Analyis—CFA

The results of the CFA supported the use of a three-factor model to identify the critical success factors for BIM implementation. Table 3 lists the CFA results, in which all of the standardized coefficients are higher than 0.5. Goodness of fit (GOF) was proposed to evaluate the model. The results were satisfactory, and a value of $\chi^2/df = 2.90 < 3.00$ was obtained. The comparative fit index (CFI) was 0.96, the root mean square error of approximation (RMSEA) was 0.04, and the goodness of fit index (GFI) was 0.97. These results indicate a good model fit. The fitness ratios of the model indicate that the CFA model fits the data well. Therefore, the model allows the verification of the measurement scales. All CSC and latent factors included in the hypothetical model were thus considered reliable, and SEM was utilized to test the theoretical model.

### 4.5. Evaluation of Hypothetical Model

After the CFA checked the validity of the measurement scale, a hypothetical model was constructed and three hypotheses were developed (Figure 3). Each path in the model presents a hypothetical relationship between a pair of constructs.

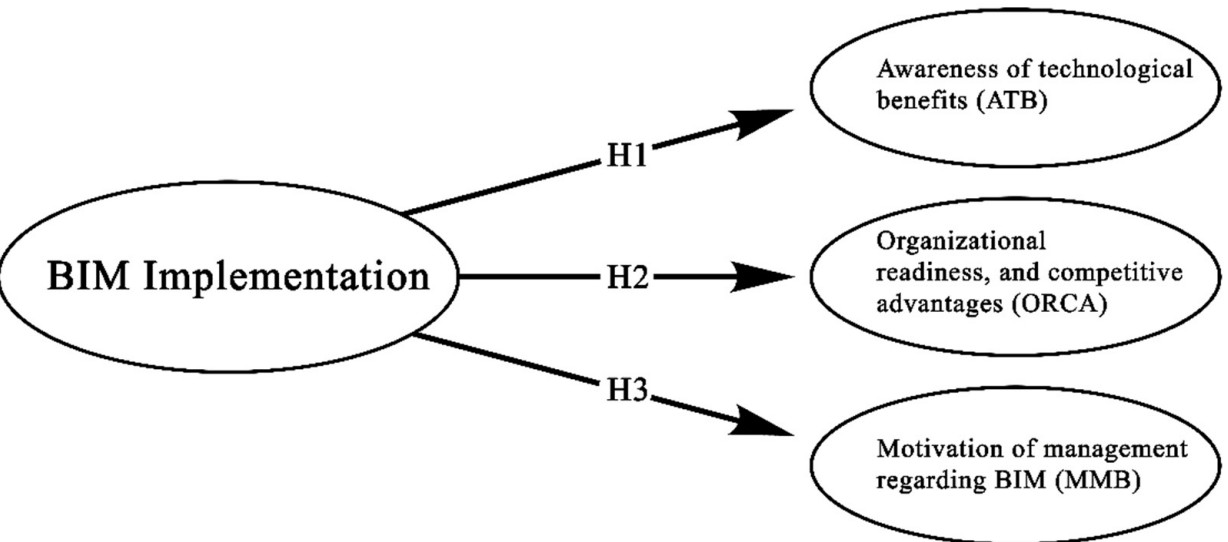

**Figure 3.** Hypothetical model of critical success factors for BIM implementation.

Regarding the three latent factors (ATB, ORCA, and MMB), critical success factors for BIM implementation were considered. Therefore, three hypotheses were established (paths in Figure 3).

**H1.** *ATB has a direct, positive effect on BIM implementation.*

**H2.** *ORCA has a direct, positive effect on BIM implementation.*

**H3.** *MMB has a direct, positive effect on BIM implementation.*

#### 4.5.1. Reliability Testing

Regarding the conceptual model (Figure 3), SEM was developed (Figure 4), which presents the standardized path coefficients of each hypothesis. The measurement model's convergent validity and individual item reliability values, which were concurrently produced, must be evaluated.

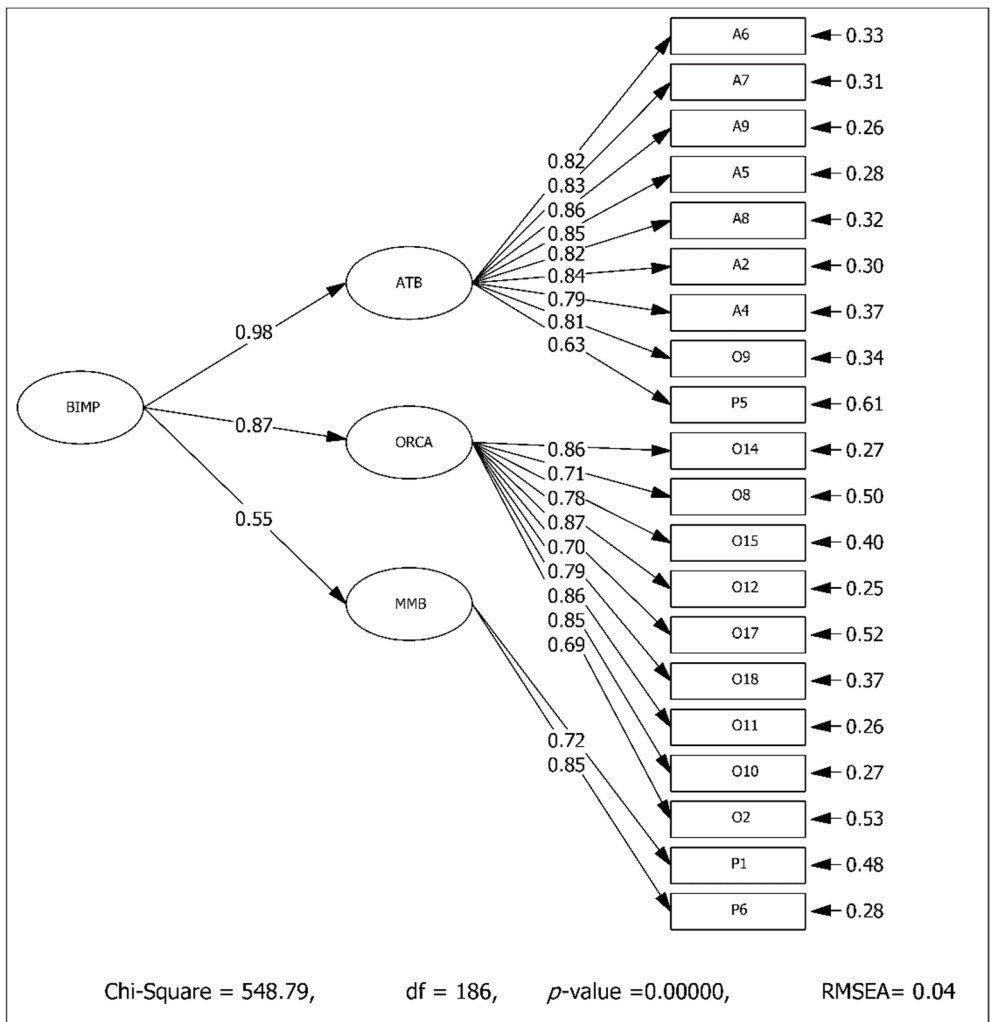

**Figure 4.** Proposed latent variable model.

The construct's reliability can be quantified using two methods: Cronbach's alpha (α) (CA) and composite reliability (CR) [158]. Because the indicators are not equally reliable, the rule of thumb for both reliability criteria is that they must be greater than 0.70. Because CR (weighted) is more accurate than CA (unweighted), it should be evaluated and reported. Both CA and CR assessed the reliability of the model constructs in this study. The degree to which items regularly display the hidden construct is referred to as the construct reliability [159]. In the present study, the CR for all constructs/latent variables exceeded the recommended cut-off of 0.70 and ranged from 0.76 to 0.94, and all CA values surpassed the recommended cut-off of 0.70 and ranged from 0.75 to 0.94, as listed in Table 4.

**Table 4.** Reliability results and AVE results of constructs/latent factors.

| Constructs/Latent Variables | CR | Cronbach's Alpha (CA) | AVE |
|---|---|---|---|
| ATB | 0.94 | 0.94 | 0.65 |
| ORCA | 0.93 | 0.93 | 0.63 |
| MMB | 0.76 | 0.75 | 0.62 |

4.5.2. Validity Testing

The data were validated after review for reliability. The average variance extracted (AVE) test [160] was used to evaluate the internal consistency of the construct by assessing the amount of variance captured by a latent variable from its measurement items to the amount of variance captured by measurement errors.

According to Fornell and Larcker [160] and Hair et al. [158], the AVE should be greater than 0.5, indicating that the latent variables accounted for at least 50% of the measurement variance. In this study, all AVE values ranged from 0.62 to 0.65, exceeding the required AVE of >0.50, supporting the use of all constructs (Table 4).

*4.6. Evaluation of Structural Model*

After determining the reliability and validity of the measurement model, its structure was tested. Four distinct tests were conducted to establish the inner model, as indicated by Urbach and Ahlemann [161], and Ramayah et al. [162]: the coefficient of determination ($R^2$), GOF measures, the *t*-value test, and the path coefficient.

The model was assessed based on the first GOF measures, as shown in Table 5.

**Table 5.** Summary statistics of model fitness indices.

| Fit Index | Suggested Values | Observed Values | Evaluation |
|---|---|---|---|
| $\chi^2/df$ | $\chi^2/df \leq 3$ | 2.95 | Excellent |
| GFI | $0.95 \leq GFI \leq 1.00$ | 0.98 | Excellent |
| AGFI | $0.90 \leq AGFI \leq 1.00$ | 0.94 | Excellent |
| NFI | $0.95 \leq NFI \leq 1.00$ | 0.97 | Excellent |
| CFI | $0.95 \leq CFI \leq 1.00$ | 0.97 | Excellent |
| RMSEA | $0 \leq RMSEA \leq 0.05$ | 0.04 | Excellent |

Meanwhile, the *t*-value test, path coefficient values, and $R^2$ values were also used for evaluating the results as presented in Table 6.

**Table 6.** Standardized coefficient estimates of the model.

| Hypothetical Paths and Expected Influences | Path Coefficient * | *t*-Value | $R^2$ | Interpretation |
|---|---|---|---|---|
| H1: ATB→BIMP | 0.98 | 13.36 | 0.96 | Supported |
| H2: ORCA→BIMP | 0.87 | 12.59 | 0.76 | Supported |
| H3: MMB→BIMP | 0.55 | 6.14 | 0.30 | Supported |

Note: * All standardized path coefficient estimates are expected to be significant at $p < 0.01$. ATB: Awareness of technological benefits; ORCA: Organizational readiness and competitive advantages; MMB: Motivation of management regarding BIM; BIMP: BIM implementation.

The coefficient of determination, often known as $R^2$, is a crucial measure for evaluating the structural model. The $R^2$ value indicates how much variance in the endogenous variable can be explained by one or more exogenous factors (s). The $R^2$ values, which reflect the ability of the exogenous variables to explain the endogenous variables, determine the structural model's quality. $R^2$ levels exceeding 0.67 are regarded as strong, and values between 0.33 and 0.67 are considered moderate, while $R^2$ values between 0.19 and 0.33 are considered weak, and $R^2$ values less than 0.19 are deemed unacceptable [161]. As presented in Table 6, the $R^2$ values of the current study range from 0.30 to 0.96.

The SEM revealed that all hypotheses in the conceptual model were supported. Table 6 presents the t values for each hypothesis, which exceed the critical two-tailed t value of 2.58 at the 0.01 significance level. The fully evaluated model is illustrated in Figure 4.

## 5. Discussion

The current study examined the critical success factors of BIM implementation in the Turkish AEC industry in terms of the use of BIM by architects and engineers using data from 243 questionnaire responses from those in the industry who work in companies that use BIM technology.

From the outset, the systematic literature review presented 41 SC for BIM implementation categorized under BIM practice, BIM awareness, BIM organization, and BIM education, but only 20 out of 41 CSs were determined as critical success criteria. While this does not imply that the other 21 criteria do not have any importance in term of successful BIM implementation, their impacts may not be adequate to be deemed as "critical" for

this study [38]. None of the seven criteria under the BIM education category have been identified as critical [33], which is partially consistent with the research of Evans et al., who have addressed this issue within the context of training and have found that BIM training is among the five least significant CSFs for BIM and lean practices in developed and developing countries. Some previous studies [23,28,34,75,86,163] have addressed this issue within the context of training, while others have done so [32,120] within the context of education. Unlike this study, some of them identified the issues of BIM education or training, including CSF categories [28,32,75], critical barriers [86,163], and key drivers [120]. In addition, Abbasnejad et al. [34], who investigated enablers with the SLR method for BIM adoption and implementation in their study, found that training is one of the most cited key enablers for successful BIM implementation.

Furthermore, Ozorhon and Karahan [23] identified that training of employees in BIM is one of the important factors for BIM implementation in Turkey, which is different to the findings from this study. In addition, Tan and Gumusburun Ayalp [89] found that the lack of BIM education and training opportunities was one of the root factors limiting BIM implementation in their study. It is thought that the main cause of this difference is the sampling groups of these studies. That is, the sample group of this study consists of architect and engineers in companies in which BIM is implemented, while the sample group of Tan and Gumusburun Ayalp [89] in their study is architects and engineers who do not use BIM technology. Therefore, it can be concluded that the problems related to BIM education have begun to be solved by BIM user construction firms.

The EFA and CFA revealed three critical success factors that affect the success of BIM implementation. With the construction of SEM, the critical success factors of BIM implementation were identified.

SEM supported all three hypotheses. As shown in Figure 4 and Table 6, there are differences in the path coefficient values of the hypotheses. Path coefficient values that are approximately one imply a strong association, whereas values near zero indicate a weak relationship [164]. Therefore, the significance and effect of the critical success factors could be divided into two groups: those with a path coefficient range between 1 and 0.81 are the most crucial and have a very high effect, and those between 0.60 and 0.40 have a moderate effect. While this does not imply that a critical success factor that is determined to be moderate does not have any significance with regard to BIM implementation, it will guide the determination of priority issues that need to be resolved.

Based on the path coefficient groupings, two critical success factors occur in the first group. "Awareness of technological benefits (ATB)" is the most influential critical success factor of BIM implementation with a path coefficient of 0.98. Although previous studies [40,66] identified this factor under the list of the top drivers of BIM implementation in different developing countries, this study revealed that "awareness of technological benefits" is the most important key success factor for BIM implementation. In addition, the result regarding awareness of technology is partially consistent with the research of Babatunde et al. [36], which identified awareness of technology as a criterion only under the list of BIM adoption drivers. Along with this, the lack of awareness of BIM and of BIM potential is a core factor affecting BIM implementation and BIM maturity [71,89].

The second most important key success factor is "Organizational readiness and competitive advantages (ORCA)", with a path coefficient of 0.87, which is as significant as that of ATB. This result is partially consistent with the research of Chen et al. [26], who found that organizational readiness has an important direct effect on BIM adoption, with a path coefficient of 0.349 in sample group 1 of their study, which was directed toward engineering consulting firms. Organizational readiness is a critical factor for technological adoption in businesses [20] and is related to whether the organization has the knowledge, skills, expertise [23], and available resources for technology adoption. Moreover, the readiness of the organization for change and its ability to manage change are critical factors that facilitate BIM implementation [165]. In addition, this result is partially consistent with the

research of Phang et al. [117] who identified that using BIM as a competitive advantage is one of the eight critical success factors.

Among the moderately influential critical success factors contained within the second group, "Motivation of management regarding BIM (MMB)", a path coefficient of 0.55 emerged, which was relatively high. One of the two criteria that constitutes the content of this success factor, the reduction in cost and time by BIM technology, is highlighted in most recent studies as a driving factor [36,64], and as a benefit [24,37]. Furthermore, according to Malik et al. [128], the savings in both time and cost are among the reasons for the increasing interest in the implementation of BIM in the construction industry. Previous studies [20,59,89] highlighted the significant role of management support. Nonetheless, management motivation is an important cause underlying management support. Profit is the core motivational power for innovation in the construction industry [166]. The ease of learning BIM-based programs by employees reduces cost and time, and increases the motivation of the management, as they provide fast implementation and profit. Thus, it could be stated that MMB is an important factor for successful BIM implementation. Unlike previous research, this study revealed the original success factor of the motivation of management regarding BIM. The defining factors of BIM implementation and BIM practices are shaped by the industry, so there is a need to conduct research in the natural environment of an organization or a country, depending on industry stakeholders [167,168]. Therefore, environmental and contextual dissimilarities may explain this differentiation in the findings obtained.

## 6. Conclusions

BIM technology has significant advantages for all construction stakeholders in the AEC industry. Although the implementation of BIM is at a low level in the Turkish construction industry, it is known that minor construction companies have started this practice in a manner similar to other developing countries' AEC industries. To promote BIM implementation in developing countries, the critical success factors of BIM implementation should be disseminated. Therefore, this study identifies the critical success factors of BIM implementation using SEM, achieved by sampling construction professionals who use BIM, a group that has been ignored in previous research.

Initially, the systematic literature review presented 41 SC for BIM implementation categorized under BIM practice, BIM awareness, BIM organization, and BIM education. Using 41 SC, a questionnaire was organized, and data collection was performed in Turkey. The participants (architects and civil engineers) were selected from the construction industry among those construction firms that implement BIM technology. A total of 243 fully completed survey forms were statistically analyzed, and 20 out of 41 CSs were determined as critical success criteria. To examine the main critical success factors (CSFs), an exploratory factor analysis was conducted, and three CSFs were extracted. As modeling the critical success factors of BIM implementation was one of the main aims of this study, structural equation modeling (SEM) was conducted for these three CSFs. SEM revealed that awareness of technological benefits (ATB), organizational readiness and competitive advantages (ORCA), and motivation of management regarding BIM (MMB) were the critical success factors for BIM implementation, with path coefficients of 0.98, 0.87, and 0.55, respectively.

This study is unique in that it assists construction researchers and stakeholders in specifying the critical success factors that enable efficient BIM implementation, which is critical for BIM implementation throughout a project's lifetime. Furthermore, this study is the one of the limited number of studies to have developed a quantified model to demonstrate and measure the effect size of the critical success factors of BIM implementation, which will be useful for policymakers and organization managers in Turkey to devise appropriate frameworks for BIM adoption in the Turkish AEC industry. Additionally, these frameworks could be generalized to other developing countries. This model is therefore valuable for construction stakeholders and firms to adopt BIM in their projects. Furthermore, this study

has crucial management implications and empirical contributions for the AEC sector, which are outlined below.

### 6.1. Conceptual and Empirical Contributions

The proposed model proved the need for successful BIM implementation, particularly in the AEC industries of developing countries. This study determined the achievement criteria of BIM implementation using the proposed model. As a consequent, the gap between BIM theory and practice is closing. To the best of our knowledge, no study has investigated the critical factors of BIM implementation in the Turkish AEC industry, which was achieved by sampling construction professionals who use BIM. In the first stage of the study, SC were identified using the SLR. This result paves the way for further research on the factors that drive successful BIM implementation in developing countries, especially in the field of construction management. To this end, the theoretical aspects of this study provide a mathematical framework for identifying and quantifying the critical factors for the achievement of BIM implementation in Turkey and other developing countries. Using SEM, three factors that demonstrate successful BIM implementation in the Turkish construction industry were determined. As a result, this study provides a framework for policymakers who are interns to incorporate BIM impartially. Furthermore, this study made several conceptual and empirical advances which are as follows:

- The study contributes to the conceptual framework by identifying and conceptualizing additional components to be added to it, such as the influence of critical success factors on BIM implementation in the construction industry.
- Most BIM implementation studies have been conducted in developed countries (the United States of America, the United Kingdom, France, and South Korea). Few studies have been conducted on BIM implementation in the Turkish construction industry; therefore, the findings of the current research may be generalized to developing countries. The present work lays a solid foundation for addressing BIM implementation to increase the dependability of local construction projects and to close the knowledge gap highlighted earlier.
- This study's findings include a substantial prediction tool (SEM) for discussing the influence of successful BIM implementation factors on the AEC industry for the first time, by sampling a group that has been ignored in previous research. Consequently, this tool has the potential to improve traditional BIM implementation in the construction industry, particularly in developing countries. This contribution is empirical in nature because, in terms of the used method and sample group, it focused on evaluating a theoretical linkage between three critical factors of successful BIM implementation, which have not previously been tested.
- Regarding the geographical context, it is evident that BIM implementation remains at a low level, with minor companies having just started to adopt this practice, similar to other developing countries' AEC industries; this is projected to skyrocket over the next few years. The present empirical study provides evidence that there is a significant and positive impact of the critical success factors of BIM implementation in the construction industry. Consequently, this can encourage the Turkish government and other local organizations to implement BIM more successfully, and it can also encourage those who have not yet started to implement BIM.

### 6.2. Managerial Implications

The following managerial implications can be used by construction practitioners to understand the impact of critical success factors for BIM implementation.

Accordingly, "Awareness of technological benefits (ATB)" is the most influential critical factor for successful BIM implementation and thus is worthy of our attention. The following several practical recommendations can be presented to enhance the affirmative effects of this factor:

- Regarding ATB, this study suggests that BIM product developers could increase the ratio of utilization of BIM technology by determining what type of implementations should be used in an organization, and arranging more BIM seminars, workshops, or conferences for key stakeholders in this industry, such as top management, clients, chief contractors, and engineering companies.
- The government may consider whether it is feasible to develop grant incentives and financial subsidies to local AEC industries' early phases of BIM implementation as well as consider selecting a few pilot projects in order to practically present the affirmative impacts of BIM. Furthermore, the government should consider BIM consultancy support, during any project phase that preferred by the company, for a limited period.
- In addition, both universities and the government should take responsibility for the awareness of BIM-related technologies. Related departments of universities, particularly architecture and engineering departments, should create new course modules by update their education curricula and increasing the number of BIM-related courses. Institutions of higher education and the government should collaborate to develop a national BIM curriculum to fulfil the need for BIM proficiency among graduates entering the workforce in the next generation.
- The government should implement legislation and rules during their project execution to successful BIM via providing project standardization. The government should also provide a more enabling environment for BIM users, including intellectual property protection, BIM rules, and a common contract for BIM implementation, thus decreasing project risk to a minimal level.

The second most important key success factor is "Organizational readiness and competitive advantages (ORCA)" which enhances the affirmative effects of this factor.

- The government should fund projects in order to build roadmaps of organizational readiness for BIM implementation for each firm as needed. Thus, it can facilitate organizations to overcome their shortcomings in order to implement BIM more successfully.
- The government should establish educational programs for current industry professionals and create a knowledge portal to evaluate the most effective BIM implementation approaches by via strategic schemes. Furthermore, construction stakeholders for successful digital transformation should motivate their staff to acquire practical skills and real-world experience through frequent BIM training programs that help them to stay updated and be aware of abilities that need to be improved.
- In addition to the organizational readiness process for successful BIM implementation, government subsidies have positively affected companies' attitudes towards change by provide long-term competitive advantages.

Accordingly, "Motivation of management regarding BIM (MMB)" is the last influential critical factor for successful BIM implementation to enhance the affirmative effects of this factor:

- Since the gains in design cost and time have significant influence on the motivation of the top management to implement BIM, local companies in the AEC industry should be encouraged to implement BIM by the government via sharing the results of the pilot projects.
- It should be noted that BIM implementation is not only technological innovation, but it also requires the digital transformation of the entire organization. Organizations would have better results if companies focused on this transformation and were aware of the need to transform digitally. The government should organize informative seminars and conferences on process management to raise awareness. In line with awareness, knowing reliable results will be obtained with the correct implementation is an important source of motivation.
- Effective communication between executives and workers has a significant role for successful BIM implementation in term of better involvement of the workers in the digital transformation process and in achieving the objectives related to implementation. Therefore, various workshops should be organized to consolidate/strengthen

the linkage of executives and workers, and to ensure success in BIM implementation in terms of affirmative feedback.

The suggested conceptual framework revealing the critical success factor groups is unique and can support those implementing BIM in identifying roadmaps with specific modifications to their BIM implementation operations to improve the efficacy of BIM implementation and to provide achievement. Consequently, the primary results of this work contribute to BIM implementation scholarship. To enhance the impacts of the critical factors of successful BIM implementation, project teams, which include executives and workers from significant stakeholders, should adapt their usual work practices.

## 7. Limitations and Future Research

Despite the great efforts made in this study to contribute significantly to the determination of the critical success factors of BIM implementation, it has some limitations. These limitations will lead future studies. First, it was limited in terms of geographical location. The research questionnaire was administered only to architects and engineers who worked at organizations that implemented BIM in Turkey. Future studies should seek to further explore other developing countries to improve the generalizability of the results, and to build upon this work with new contributions. Besides architects and engineers, different stakeholders (contractors, developers and material suppliers) should be incorporated into the sampling. By these means, the perception of SC and CSFs of BIM implementation among various stakeholders could be examined in future studies. This study contributes to identifying the critical success factors of BIM implementation in the Turkish AEC industry by using SEM with theoretical conceptualization. Innovation diffusion theory or roadmaps may be used to disseminate BIM implementation and to develop a strategy for increasing BIM implementation in developing countries. Moreover, the developed SEM provided a good fit. Future research could focus on the mediating influence of organization culture or the demographic variables of participants.

**Author Contributions:** S.T., G.G.A., M.Z.T., M.S. and Y.B.M. contributed to all stages of the study, from the conceptualization to the preparation of the article. S.T. and G.G.A. contributed to conceptualization. S.T., M.S., M.Z.T. and Y.B.M. contributed systematic literature review stage. S.T., M.Z.T., M.S. and Y.B.M. contributed to the data collection. G.G.A. contributed qualitative data analysis stages. S.T., G.G.A., M.S., M.Z.T. and Y.B.M. contributed to writing, editing, and reviewing the manuscript. All authors have read and agreed to the published version of the manuscript.

**Funding:** The research received no funding.

**Institutional Review Board Statement:** Not applicable.

**Informed Consent Statement:** Informed consent was obtained from all subjects involved in the study.

**Data Availability Statement:** The data are available upon request.

**Conflicts of Interest:** The authors declare no conflict of interest.

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
