# Peer review of "Modeling the Critical Success Factors for BIM Implementation in Developing Countries: Sampling the Turkish AEC Industry"

_sustainability, doi:10.3390/su14159537_

Round 1

Reviewer 1 Report

I have reviewed the manuscript and come up with the following comments:

Introduction:

The section is not well structured (lack of logical sequence in the content), which also does not clearly provide a basis for the research aim. I also noticed many useless information, which has almost no relationship with the research aim. 

Based on the SLR, it seems like there are many reported on the subject. The author(s) need to clearly differentiate the current study from those that are reported in the literature. 

The research methodology has been clearly presented. However, I strongly recommend to focus on the discussion section and ensure that it pulls out the most significant outcomes of the research. The current version looks more like a results section with some support from the literature, which does not make it robust. 

The research and practical implications are missing and have to be clearly discussed. Lastly, the manuscript has to be professionally proofread. 

Author Response

Authors appreciate to reviewer’s valuable contribution. We revised the manuscript based on the comments and suggestions. Additionally, all revised parts were indicated in red colour.

Reviewer 2 Report

The paper presents Modeling the Critical Success Factors for BIM Implementation in Developing Countries: Sampling the Turkish AEC Industry. It is a well structured and designated paper by using questionnaire survey for the investigation.

A) But the paper needs to clarify its research reliability on:

1) The key bias to this paper is that it has not clearly stated the sampling method that is used for the questionnaire in Section 3.3 Data collection (Page 11) and why the sampling method is appropriate to eliminate bias, which affects the reliability of the questionnaire and the paper.

2) The Table for The Cronbachs α coefficient of the dataset for the 41 success criteria (SCs) regarding 464 BIM implementation needs to be provided in Section 4.1. Reliability and validity of questionnaire.

B) However, the paper is a purely general BIM adoption questionnaire study without considering any issues regarding (directly or in-directly) Sustainability and its special issue “Digital Sustainability in Building Design”, which is out of the scope of Sustainability https://www.mdpi.com/journal/sustainability/about   and its special issue “Digital Sustainability in Building Design” https://www.mdpi.com/journal/sustainability/special_issues/digital_building 

Author Response

(The authors gave the same response as above.)

Reviewer 3 Report

This type of investigative studies should be applauded for the best implementation of a more efficient Methodology in the AEC sector, such as BIM, constituting part of the economic engine of any country. Therefore, this study is suitable to identify and model the critical success factors for BIM implementation, in order to gain insights for fast and efficient BIM implementation among construction companies in the Turkish AEC industry.

 The design of the questionnaires with 41 identified success criteria (41 CS) are successful and are supported through a systematic review of the literature (SLR).

 The survey was conducted among construction professionals who actively implement BIM technology in their busy companies in Turkey architects and civil engineers.

 Exploratory Factor Analysis (EFA) is successful in identifying Critical Success Factors (CSFs) as well as Structural Equation Modeling (SEM) in examining the underlying size effects of each CSF in BIM implementation.

 The three critical success factors for the implementation of BIM in the Turkish construction industry: G1.- “Awareness of technological benefits” and “Organizational readiness and competitive advantages”; and G2.-. “Management Motivation for BIM” are well supported and logically appropriate factors. 

Only the following consideration should be taken into account: The current study examined the critical success factors of BIM implementation in the Turkish AEC industry only by interviewing architects and engineers, without considering other important actors, such as developers. 

The document should be reinforced in terms of the established limitations of the people interviewed. In addition to architects and engineers, different stakeholders (contractors and material suppliers) should be included in the sampling. 

In this way, it could be affirmed that the study contributes to identify the critical success factors of the implementation of BIM in the Turkish AEC industry through the use of SEM with theoretical conceptualization.

Author Response

We sincerely thank you for your kind evaluation. If we had interviewed the participants, it would have been possible to include contractors, material suppliers and developers as participants. However, we used questionnaire as data collection instrument and collected the data from February to May 2022. Therefore, future studies may focus on contractors, developers and material suppliers or handled the issue from the all construction stakeholders’’ perspectives. In addition, your suggestion for sampling method as including developers, is added as limitation the current research which may be addressed at future researches. 

Round 2

Reviewer 1 Report

The authors have addressed all my comments. The manuscript is now suitable for publication. 

Author Response

Thank you for your kind evaluation

Reviewer 2 Report

The paper has been improved with one issue left that needs to be further clarified:

In terms of sampling method, Section 3.3 Data collection (Page 13), the random sampling method has claimed to be used, however the specific population pool (total population) for sampling has not been supplied, since this research is not study mass population, which is the key for the using questionnaire survey for the investigation.

Author Response

Thank you for your kind evaluation. Based on your comments,authors indicated the total population and its reference at Page 15 Line 525-527 in red color as “The population number in this research is extracted from the statistics produced by Turkish Chamber of Engineers and Architects Union for 2022 as 205843 [140], which states the number architects and civil engineers of total 68478 and 137365 respectively.”

Due to adding a new reference numbered as “[140]” in new version, the refence numbers after 140 are changed. Authors corrected them also in manuscript and reference list.
